# Proposal of Nutritional Standards for the Assessment of the Nutritional Status of Grapevines in Subtropical and Temperate Regions

**DOI:** 10.3390/plants14050698

**Published:** 2025-02-24

**Authors:** Danilo Eduardo Rozane, Moreno Toselli, Gustavo Brunetto, Elena Baldi, William Natale, Betania Vahl de Paula, Juliana Domingues Lima, Fabiana Campos Medeiros, Gustavo Ayres, Samuel Francisco Gobi

**Affiliations:** 1Department of Agronomy and Natural Resources, São Paulo State University (UNESP), Registro 11900-000, SP, Brazil; danilo.rozane@unesp.br (D.E.R.); juliana.d.lima@unesp.br (J.D.L.); 2Dipartimento di Scienze e Tecnologie Agro-Alimentari, Università di Bologna (UNIBO), 40126 Bologna, Italy; moreno.toselli@unibo.it (M.T.); elena.baldi7@unibo.it (E.B.); 3Department of Soil Sciences, Federal University of Santa Maria (UFSM), Santa Maria 97105-900, RS, Brazil; william.natale@ufsm.br (W.N.); behdepaula@hotmail.com (B.V.d.P.); 4Postgraduate Program in Agronomy, São Paulo State University (UNESP), Jaboticabal 14884-900, SP, Brazil; fabiana.medeiros@unesp.br; 5Postgraduate Program in Environment and Sustainability, State University of Rio Grande do Sul (UERGS), São Francisco de Paula 95400-000, RS, Brazil; gustavo.ayres@gmail.com; 6Department of Agronomy, University of the Campanha Region, Bagé 96400-000, RS, Brazil; samuel-gobi@hotmail.com

**Keywords:** leaf analysis, leaf abscission, CND, discriminant analysis, fertilization, *Vitis vinifera*, *Vitis labrusca*

## Abstract

The necessity for nutritional standards to evaluate the nutritional status of grapevines is a critical concern for viticulturists worldwide. This study addressed the lack of multinutrient standards that consider specific genetic and environmental factors by proposing regional standards based on data collected under different growing conditions. Using the compositional nutrient diagnosis (CND) method and multivariate analyses, leaf samples from 585 commercial vineyards in Emilia-Romagna, Italy, and the states of São Paulo and Rio Grande do Sul, Brazil, were evaluated. The results confirmed significant variations in nutritional standards among regions and cultivars, emphasizing the need for regional adjustments in fertilization recommendations. This work proposes critical levels, sufficiency ranges, and nutritional standards that can improve grapevine nutritional assessments, promoting greater precision in fertilization management. The findings reinforce the importance of regional standards, avoiding the use of unsuitable universal recommendations.

## 1. Introduction

In 2022, the global vineyard area was estimated to be 7.3 million ha, showing a slight decrease (0.4%) compared to 2021. Five countries account for 50% of the world’s vineyards: Spain (13.1%), France (11.2%), China (10.8%), Italy (9.9%), and Turkey (5.6%). Brazil ranks in the 22nd position, with a 1.1% share of global production [1].

Effective fertilization recommendations should be based on both soil and plant tissue analyses. Such analyses are crucial for managing fertilization appropriately, particularly in fruit crops such as grapevines, which can take up nutrients from deeper soil layers [2]. Moreover, ontogenetic variation in genetic materials should also be considered [2,3]. Grapevines also have adaptive strategies that allow them to solubilize and take up nutrients, such as potassium (K), that were considered unavailable (non-exchangeable) [4]. These adaptations eventually cause variations in nutritional assessment standards, as observed in other species [2]. As a result, there is growing interest in updating nutritional standards for fruit crops, mainly in different production environments where cultivation conditions are dynamic and highly variable [3].

The assessment of nutritional status based solely on critical nutrient levels, i.e., using only univariate data, fails to account for nutrient interrelationships, which are governed by a network of physiologic processes influenced by both phenotypic and genetic factors impacting the leaf nutrient levels measured in chemical analyses [5]. Nutrient interactions occur at multiple levels, resulting not only in synergistic or antagonistic effects but also in changes due to chemical similarities between elements that are interconnected and self-adjust within the plant tissue compositional space [5,6]. Consequently, nutritional assessments performed on plant tissues should be viewed as single multivariate datasets and should not be interpreted independently [5,7].

Nutritional diagnostics in viticulture have traditionally relied on univariate methods, such as critical nutrient levels, which fail to account for the complex interrelationships among nutrients. Recent advancements in multivariate techniques, including the Compositional Nutrient Diagnosis (CND) method, provide a more robust framework for evaluating plant nutrition by considering nutrient interactions within a multidimensional compositional space. These approaches have demonstrated superior precision in identifying nutritional imbalances compared to traditional methods, making them an essential tool for modern viticultural practices.

The application of CND in grapevine nutrition is particularly relevant given the crop’s sensitivity to soil and climatic variability. Studies have shown that grapevine nutrient uptake is significantly influenced by factors such as soil pH, organic matter content, and climatic conditions, which vary widely across different wine-producing regions [3,4]. Despite this knowledge, there remains a lack of region-specific nutritional standards that integrate these variables, leading to the overuse or underutilization of fertilizers in many vineyards.

The following null hypotheses are therefore stated: (1) the nutritional status of grapevines differs across regions, cultivars, and cultivation conditions, and (2) it is possible to propose nutritional standards for each of the assessed conditions. Consequently, the aim of this study was to assess nutritional similarity among grapevine cultivars across different regions and to propose multinutrient standards for each condition. This will facilitate the assessment of nutritional status and help streamline the fertilization recommendations for vineyards.

This study relies on the CND method to analyze a comprehensive dataset collected from several grape-growing regions in Brazil and Italy. By integrating multivariate statistical tools and field-specific data, this work not only proposes personalized nutritional patterns but also highlights the critical role of region- and cultivar-specific approaches in optimizing fertilization strategies. This innovative methodology could set a benchmark for future research and addresses the need for localized guidelines in viticulture.

## 2. Materials and Methods

### 2.1. Data Collection

The grapevine dataset consisted of 585 complete leaf samples collected from *Vitis vinifera* L. cultivars in different years (Table 1). The samples were obtained from commercial vineyards in Bologna, Italy (leaf blade), São Paulo (SP) and Rio Grande do Sul (RS), Brazil (leaf blade + petiole). The study included 18 grape varieties used in the production of wine and sparkling wine from five different locations: Sangiovese ‘clone Tea 10′ (Bologna); APPC 7—Estela (Pilar do Sul—SP); Moscato Branco (MB) and Bordô (B) (*Vitis labrusca* L.) (Farroupilha—RS); Tannat, Malbec, Sauvignon Blanc, Chardonnay, Gewurztraminer, Merlot, Pinotage, and Cabernet Sauvignon (Dom Pedrito—RS); Cabernet Sauvignon, Cabernet Franc, Chardonnay, Malbec, Merlot, Pinot Noir, Ruby Cabernet, Syrah, Tannat, Tempranillo, and Viognier (Maçambará—RS). The ‘Sangiovese’ vines were grafted onto rootstock 110 Richter (*V. berlandieri* × *V. riparia*), whereas the ‘APPC 7—Estela’ vines were grafted onto rootstock 420 A (*V. berlandieri* × *V. riparia*). The cultivars in Brazil were grafted onto rootstocks 1103-P (*V. berlandieri* × *V. rupestris*) and SO4 (*V. berlandieri* × *V. riparia*).

Bologna is located in the Emilia-Romagna region in Italy (44°29′54.8″ N and 11°20′43.9″ E, at an altitude of 54 m). Pilar do Sul is located in the state of São Paulo (23°48′34.5″ S and 47°43′18.8″ W, at an altitude of 690 m). Farroupilha is situated in the northeastern mesoregion of Rio Grande do Sul (29°13′25″ S and 51°20′54″ W, at an altitude of 768 m). Dom Pedrito and Maçambará are located in the southernmost part of the state of Rio Grande do Sul (30°00′22.1″ S and 55°25′38.7″ W, at an altitude of 130 m).

The evaluated plots are from adult areas with ages between 8 and 15 years of planting. Depending on production, soil properties, nutritional demands, and local indications, fertilization in Brazil was adjusted for the region of Farroupilha, Dom Pedrito and Maçambará according to the indications of [8] with variations in nitrogen fertilization at 10, 20, and 30 kg ha^−1^ of N depending on organic matter of <25; 26–50; >50 g kg^−1^, respectively, and phosphate and potassium fertilizers ranging from 20 to 60 and 20 to 50 kg ha^−1^ of P_2_O_5_ and K_2_O, respectively. Fertilization in Pilar do Sul follows the recommendations of [9] with variations in nitrogen fertilization between 100 and 150 kg ha^−1^ of N depending on the expected productivity <20; 21–35; >35 kg ha^−1^, respectively, and phosphate and potassium fertilizer ranging from 120 to 180 and 120 to 300 kg ha^−1^ of P_2_O_5_ and K_2_O, respectively. Fertilization recommendations in Bologna were carried out with the application of varying rates of organic compounds between 0 and 20 Mg kg ha^−1^ depending on production expectations.

Table A1 presents the median values of soil fertility attributes and climate variations in the experimental sites during the evaluated period. It should be noted that the values presented in the fertility attributes consider the extraction by Mehlich-1 for the concentrations of P, K, Ca, Mg, Cu, Fe, Mn, and Zn in the locations of Bologna in Italy and in Farroupilha, Dom Pedrito and Maçambará in Brazil and the Pilar do Sul region in the state of São Paulo, Brazil employs the Resina extractor to determine P, K, Ca, Mg, and DTPA for extraction of Cu, Fe, Mn, and Zn, and in all locations the H_2_O-Hot extractor is used for B and S-SO_4_ determined with calcium phosphate by turbidimetry.

### 2.2. Evaluations and Tissue Analyses

Sangiovese leaves were collected in summer (Sum) and at abscission (Abs). With the exception of Abs, leaves from all genotypes were collected from the middle third of the annual shoot at bloom. Leaves were dried in a forced air oven at 60 °C ± 5 °C until they reached a constant mass and then ground to pass through a 0.841 mm (20 mesh) sieve. A subsample was digested using sulfuric acid and analyzed for N using the micro-Kjeldahl method. After the digestion of the sample in a mixture of nitric and perchloric acids (2:1), K, Ca, Mg, Cu, Fe, Mn, and Zn were quantified by atomic absorption spectrophotometry, P and B by colorimetry, and S by turbidimetry, as described in [10]. Metal concentrations, in leaf samples collected in Bologna, were determined by a plasma spectrometer (ICP-OES; Ametek Spectro, Arcos, Kleve, Germany) after wet mineralization by treating 0.3 g of leaves in an Ethos TC microwave lab station (Milestone, Bergamo, Italy) with 8 mL of HNO_3_ (65%) and 2 mL of H_2_O_2_ (30%), at 180 °C for 20 min.

### 2.3. Calculations and Statistical Analyses

Calculations were carried out separately for each cultivar. In the case of Sangiovese, each collection period was taken into account. The cultivars from Dom Pedrito and Maçambará were grouped by municipality because of the limited individual representation of each cultivar. The results for leaf nutrients obtained separately from each of the seven datasets were analyzed using the calculations earlier proposed [7,11]. In addition, we point out that the compositional nutrient diagnosis (CND), with the technique centered log ratio (clr), therefore (CND-clr) transformation, following the methodology of [12], considers all nutrients and generates multinutrient variables, each weighted by the geometric mean of the nutrient composition [13].

The representation of CND-clr approach for the compositional data are shown in Equation (1):(1)$${clr}_{j}=ln\frac{x_{j}}{g\left(x\right)},$$
where $x_{j}$ is the *j*-th component in the numerator, j = [1... D], and $g\left(x\right)$ is the geometric mean of all components (Equation (2)):(2)$$g={\left(AxBx\ldots xnxR\right)}^{\frac{1}{n}},$$
where g is the geometric mean of the dry matter nutrient concentrations, adapted from [11]. The component *R* was calculated using Equation (3):(3)$$R=1,000,000-\left(A+B+\ldots +n\right),$$
where 1,000,000 is the total amount in mg kg^−1^. The sum of the nutrients in mg is subtracted from this value.

After the establishment of the multinutrient variables, the Mahalanobis distance (*MD*) [14] was calculated to exclude outliers, using the covariance matrix (CM), based on the average of the CND-clr transformations (Equation (4)):(4)$$MD=\sqrt{{\left({clr}_{j}-{clr}_{j}^{*}\right)}^{T}{COV}^{-1}\left({clr}_{j}-{clr}_{j}^{*}\right)},$$
where ${clr}_{j}$ is the sample to be compared, ${clr}_{j}^{*}$ is the arithmetic mean of the reference population, and CM is the covariance matrix of the reference population. The R component is excluded to avoid uniqueness.

Compositional nutrient diagnosis standards use means and standard deviations, which correspond to the VX relations of the centered logarithmic transformation of d nutrients from high-yield specimens, i.e., $V_{N}^{*},V_{P}^{*},V_{K}^{*},\ldots {\;V}_{R}^{*},$ and ${SD}_{N}^{*},{SD}_{P}^{*},{SD}_{K}^{*},\ldots {\;SD}_{R}^{*}$, respectively. The CND indices for d elements were calculated, as shown in Equation (5):(5)$$IN=\frac{\left(VN-V_{N}^{*}\right)}{{SD}_{N}^{*}},IP=\frac{\left(VP-V_{P}^{*}\right)}{{SD}_{P}^{*}},IK=\frac{\left(VK-V_{K}^{*}\right)}{{SD}_{K}^{*}};\ldots IR\frac{\left(VR-V_{R}^{*}\right)}{{SD}_{R}^{*}},$$
where $V_{X}^{*}$ and, $S_{X}^{*}$ are the mean and standard deviation of the X element in the high-yield subpopulation and $I_{X}$ is the CND index of the X element.

Independence between the data are ensured by the centered logarithmic transformation [12]. The CND indices were normalized, and variables were adjusted to be linear in various dimensional contexts: a circle (d + 1 = 2), a sphere (d + 1 = 3), or a hypersphere (d + 1 > 3), within a d +1 dimensional space. The nutritional imbalance index *r*^2^ was distributed as the variable $X_{d}^{2}$ with the CND indices representing independently reduced variables, as shown in Equation (6):(6)$$CND{-r}^{2}=I^{2}N+I^{2}P+I^{2}K+\ldots +I^{2}R.$$

The radius *r* is computed in the *CND* nutrient index to represent each sample based on the global imbalance *CND*-*r*^2^.

The reference population provides the parameters $V_{X}^{*}$ and $S_{X}^{*}$. A sampling error occurs because only one subgroup of the population of interest is included. Hence, confidence intervals are computed for the upper and lower bounds of $I_{N}^{*},{\;I}_{P}^{*},{\;I}_{K}^{*},\ldots I_{R}^{*},$ based on the multinutrient standards, as shown in Equation (7):(7)$$L_{R}={\bar{X}}_{R}\frac{UCB\;+}{LCB\;-}t\frac{S_{r}}{\sqrt{n}},$$
where $L_{R}$ denotes the limit for nutrient R; ${\bar{X}}_{R}$ is the average for the high-yield population for nutrient *R*; UCB is the upper confidence bound for index nutrient *R*, and LCB is the lower confidence bound for index nutrient *R*; t is the value from Student’s *t* distribution at a 5% significance level; $S_{r}$: is the standard deviation for nutrient *R*; and *n* denotes the sample size.

Assuming that the sample population is unknown, the confidence interval is understood as the likelihood that the interval between X (lower bound) and Y (upper bound) contains the true value of the population parameter. Practically, this allows us to determine the acceptable variation around the equilibrium point for $I_{N}^{*},{\;I}_{P}^{*},I_{K}^{*},\ldots {\;I}_{R}^{*}$.

By linking the nutrient concentrations in the tissue with the nutrient indices in the dataset, equations were established for each nutrient, cultivar, and region. By setting the nutrient index value to zero, it was possible to determine the critical level (CL) of the nutrient in the tissue, thereby indicating its balance. The upper (IS+) and lower (Ii−) limits of the sufficiency range were determined by applying ±2/3 of the standard deviation to the critical level value of the nutrient contents, as indicated in some studies [3].

Statistical analyses were carried out using R-4.3.1 (R Core Team, 2023) software [15].

## 3. Results

### 3.1. Principal Component Analysis

The multinutrient variables in the dataset were analyzed conjointly using principal component analysis (Table 2) to identify the variables that could most effectively indicate similarities between the groups. The values in bold indicate the variables that best explained each of the factors. Eigenvalues represent the variance explained by each attribute. Thus, the number of components was determined by selecting those with eigenvalues greater than 1.00, as already proposed [16], which explains the inclusion of four factors in this study (Table 2).

Values with significance ≥ 0.70 correspond to eigenvectors >1 [16]. Thus, the variables highlighted in bold were those that contributed the most because of the highest significance (*p* > 0.05) among the nutrients. They explained 75.4% of the model. We further highlight that factor 1 was the most important for the study because of its highest eigenvalue and explanation of 36.9% of the variance. The variables that contributed the most were region (most relevant), cultivar, year of cultivation, and the nutrients K, Mn, Ca, Mg, B, and R (residue).

The relationship between the variables associated with Factors 1 and 2 is shown in Figure 1. Note that the variables are grouped into a single cluster based on their explanatory power, indicating they are aligned by factors.

The variables that best represented the classes are grouped by region, cultivars (cv), Mn, and K, in addition, year, Ca, and Mg are taken into consideration. The latter variables appear isolated and distant from the origin. They are highly representative of Factor 1 (region, cv, Mn, K, and year) and Factor 2 (Ca and Mg). These variables are the furthest from the origin when projected perpendicularly to this factor. The other variables are less representative because they are closer to the origin of the factorial plan.

### 3.2. Discriminant Analysis

The number of roots representing the canonical variables in each component was determined by eigenvalues ≥1.00, as proposed by [16].

Table 3 shows the roots required to classify the groups. Roots 1 and 2 account for 85% of the accumulated variance for the regions, 87% for the cultivars, and 82% for years of cultivation. Wilk‘s lambda applied to the variables reveals significant differences for all attributes, regardless of the group analyzed.

Figure 2A illustrates the distribution of all groups on the Cartesian plane, indicating that each of them has a distinct nutritional profile with respect to region, cultivar, and year of cultivation. Figure 2B provides a detailed view of the dispersion of canonical observations in Pilar do Sul and Farroupilha for the Moscato Branco (MB) and Bordô (B) cultivars. 

Given the close proximity of the groups in the discriminant analysis (Figure 2A,B) and the aim to investigate potential similarities among the three most clearly defined groups [Bologna Abs and Bologna Est]; [Dom Pedrito and Maçambará], in addition to [Pilar do Sul and Farroupilha MB and B], Table 4 shows the means for the multinutrient CND variables for these populations and the ANOVA results for some nutrients (in bold). The multinutrient variable Zn was similar across Bologna, Dom Pedrito, and Maçambará groups. Variables B and R were also similar within the Maçambará group. K was similar between the Dom Pedrito and Maçambará regions and between Pilar do Sul and Farroupilha MB.

The variables Ca, Mg, and S were similar between the MB and B cultivars in Farroupilha. However, the dataset refutes the application of universal nutritional standards, highlighting the need for specific standards for each region and cultivar.

### 3.3. Critical Levels, Upper Bounds, Lower Bounds, Standards, and Confidence Intervals for CND Standards for Reference Populations of Each Nutrient in Grapevine Leaves Across Different Regions

Initial and final data were identified and quantified for each dataset after excluding atypical data (outliers) using Mahalanobis distance, thus increasing the reliability of the results. This allowed establishing the coefficients of determination (R^2^) for the statistical models examining the relationships between nutrient levels and CND indices in leaves. The lowest R^2^ values were obtained for N, followed by K. This probably occurred because of the high nutritional variability of these nutrients during the leaf sampling period (flowering), which coincides with the peak of nutrient demand [17,18], with staggered supply of N and K [19,20,21] and large redistribution of these nutrients during the grapevine’s development cycle [18,19].

Table 5 shows the proposed critical levels and upper and lower bounds for nutrient levels in grapevine leaves as per cultivar and region, reflecting the different concentrations in nutritional balances (Table 1, Table 2, Table 3 and Table 4), with references to adequate nutrient levels.

### 3.4. Correlation Between CND Summer and Autumn Nutritional Standards and Yield of the Sangiovese in Emilia-Romagna

For nutritional status assessment, collection instructions should specify the physiologic state and the plant part to be examined. The observed nutritional parameters should correlate with yield, quality, and profitability. Future assessments should follow the same sampling procedures, thus allowing for the comparison of the results with the established parameters and ensuring reliable interpretation of the nutritional status. Table 6 shows the correlation of nutrient levels and multinutrient CND standards obtained from leaf blades of Sangiovese grapevines at two developmental stages in Bologna, Emilia-Romagna, in relation to yield.

## 4. Discussion

### 4.1. Cultivar, Region, and Years of Cultivation

Multinutrients K, Mn, Ca, Mg, B, and R, in addition to categorical variables, were the ones that most significantly contributed to the estimation of models for assessing variations across cultivars, regions, and years of cultivation, whereas other nutrients played a negligible role. This might have occurred because the cultivar incorporates ontogenetic variations that determine developmental and nutrient uptake characteristics specific to each genotype [2,3]. Additionally, distinct nutrient availability due to different soil classes and soil-forming factors observed in each region [3,24], and across the years of cultivation that may vary according to the harvest period [19] may also be contributing factors.

Aside from Mn, which is likely influenced by the different soil classes in each region, the availability of other nutrients may be linked to variations in the amount and frequencies of fertilization and amendments used to neutralize acidity. This influences nutrient availability in the soil and nutrient levels in the grapevines as well [3,19,24].

All multinutrient variables contributed to explaining the categorical variables (Table 2), but Ca played a major role. This could be attributed to differences in soil formation and, consequently, in pH, affecting the requirements for soil amendments, as well as the availability, uptake, accumulation, and diagnostics across different regions, cultivars, and years of cultivation. P and B also played a role in distinguishing the groups, which is consistent with the fact that their availability is closely related to soil pH variations. Phosphate concentration in the soil solution, and its subsequent availability to plants, is in balance with the phosphate adsorbed onto the surface of Fe and Al oxides and of clay minerals [25]. Moreover, phosphate has a high affinity for Ca^2+^, Mg^2+^, Al^3+^, Fe^3+^, and Fe^2+^ ions, forming low-solubility phosphates in acidic and subacidic soils with Fe^2+^, Fe^3+^, and Al^3+^, and in alkaline soils with Ca^2+^ and Mg^2+^ [24]. Boron availability is highly dependent on pH. Indeed, in acidic soils B has a high affinity with Fe and Al hydroxides [26]. In addition, boric acid is a weak acid (pK = 9.23), with a low concentration of the H_2_BO_3_^-^ anion. Therefore, as pH increases to a certain level, the rate of ionization rises, resulting in increased adsorption. Boron deficiency is more frequent in plants grown in sandy soils, which is particularly significant for crops in the Dom Pedrito and Maçambará regions in the lowland plains of the state of Rio Grande do Sul known as Campanha Gaúcha [4,8].

Nitrogen and Cu had the greatest impact on years of cultivation. The prominent role of N may be attributed to the diverse mineralization of organic matter and the subsequent availability of N for uptake, which are affected by the climatic conditions and soil properties of each region [27]. The major influence of years of cultivation on long-term treatment experiments assessing N availability in grapevines in southern Brazil was described earlier [19]. Cupper, often used as fungicide, can accumulate over time, depending largely on the annual variations in climatic conditions [2,3].

Note that K had a minor impact on the discriminant variables (Table 2). Similar multinutrient variables for B and K were observed in the Dom Pedrito and Maçambará regions, and also for K between the Pilar do Sul and Farroupilha MB groups (Table 3). The amount of K applied tends to vary across cultivars and regions because of the different concentrations of K in the soil, leaves, and berries. Some of the K applied to the soil may be fixed in clay minerals at a 2:1 ratio [4,21] and it may not be readily available for grapevine uptake in the short term. However, some of the residual K in the soil may be taken up and partitioned into the leaves. When K concentrations in the soil are high, there may be competition with elements such as Ca and Mg [28,29]. Despite the different K concentrations in the assessed vineyards and soil types, no competitive effect with Ca was observed, considering that Ca concentrations (e.g., CL, UCB, and LCB of Ca in the tissue) were adequate when compared to the regional official standards. The only exception was in southern Brazil (Farroupilha, Dom Pedrito, and Maçambará), where Ca concentrations were below the recommended values. Nevertheless [3], reported adequate Ca concentrations in this region, in line with those found in the present study, in addition to adequate Mg levels (Table 5). In fact, similarities were noted for Ca, Mg, and S (Table 3) in the Farroupilha groups between cultivars MB and B, indicating that these nutrients did not explain the variations across regions and cultivars. This could be due to the elevated pH and high concentrations of Ca in the soils in which grapevines are grown, following the recommendation to increase base saturation to 80% for this region [8]. This probably explains the similar concentrations of Ca taken up, accumulated, and diagnosed across different cultivars.

The Zn multinutrient variables did not contribute to the discriminant analysis in distinguishing between the groups [Bologna Abs and Bologna Est] and [Dom Pedrito and Maçambará]. This could be due to the similar management practices in those regions. For instance, similar pruning practices can determine the number of buds on the current year’s shoots, thereby affecting the vegetative growth of the aerial parts, yield, and nutrient requirements throughout the plant cycle [17]. Conversely, uniform amounts and frequencies of Zn-containing fertilizers can be applied to the soil or leaves without considering specific requirements or standards. This way, Zn absorption and accumulation can be evenly distributed across the grapevines, leading to similar foliar diagnosis.

The other multinutrient variables helped explain the differences across cultivars and regions, which were expected, given the diverse management practices, environmental conditions, and genetic factors. Such variations influenced the yield and nutritional composition of grapevines in the assessed groups [3,19].

We also point out that the differences in the assessed groups could be possibly related to soil characteristics and fertilizer applications that alter the availability of native nutrients. These differences determine the amounts of nutrients absorbed and found in the tissues, including the leaves, which are the main organ used for nutritional diagnosis [19]. Moreover, climatic variables, such as temperature, differ across regions [30]. Temperature affects bud break, flowering, yield, and nutrient requirements of grapevines. Cultivars may exhibit different kinetic parameters (Km, Cmin, and Vmax), which influence nutrient uptake efficiency [31].

### 4.2. Critical Levels and Limits of Nutrients in Grapevine Leaves in Different Regions

The ratios between the coefficients of determination (R^2^) of the statistical models, which correlate the nutrient levels with the CND in grapevine leaves, are in line with research on perennial crops such as citrus fruits [2], grapevines [3] and bananas [32].

In Dom Pedrito and Maçambará, where individual representation per cultivar is limited, the cultivars were grouped into a single dataset. This approach is consistent with the recommendation made [3] for grapevines, i.e., combining the data when there are insufficient observations for each cultivar, as seen in the other regions. While the use of a single nutritional standard for a group of cultivars is not recommended [2], it can be appropriate in studies that aim to establish preliminary nutritional standards when only generic indices are available, without specific recommendations for a given cultivar and region usually provided in official manuals [8,9], or in other publications [23].

The critical levels, upper bound and lower bound, for referencing nutritional concentrations were established with yields ranging from 21.1 to 53.2 t ha^−1^ and from 30.4 to 53.2 t ha^−1^ for populations in the Bologna Abs and Bologna Est sampling time, respectively. These yields are higher than the average of 30 to 35 t ha^−1^ for the Sangiovese cultivar in the Bologna region [24].

In Brazil, the APPC7-Estela cultivar in Pilar do Sul yielded 13.0 t ha^−1^ to 33.0 t ha^−1^, which is in the average range of 20–35 t ha^−1^ recommended for this cultivar in the state of São Paulo [9]. Dom Pedrito and Maçambará, in the state of Rio Grande do Sul, showed adequate nutritional levels with yields ranging from 37.9 t ha^−1^ to 45.9 t ha^−1^ and from 5.7 t ha^−1^ to 16.7 t ha^−1^, respectively. Farroupilha, in the same state, yielded 26.0 t ha^−1^ to 81.6 t ha^−1^ for the MB cultivar and 22.9 t ha^−1^ to 53.5 t ha^−1^ for the Bordô cultivar. Except for Maçambará, all other regions surpass the maximum recommended yield for the state, which is 25 t ha^−1^ [8].

In general, normal ranges, between the LCB and UCB, are narrower than those suggested in the literature, indicating greater accuracy and precision in interpretation. They are obtained from commercial regions and consistent with the current production practices. This is probably due to the fact that the levels indicated in official Brazilian bulletins for the assessed production regions [8,9] are generic and include *Vitis vinifera* and U.S. cultivars. Likewise, in Italy [22,23], there are no culivar-specific recommendations. Therefore, one can say that the assessed datasets adjust non-specific standards and reject universally fixed and generic standards.

Sangiovese, despite the distinct variation across the observation periods (Figure 2A) in the present study (Table 5), showed ranges lower than those recommended by [22,23], suggesting that the reference values in the literature for grapevines in Italy are overestimated. An exception is Fe in Bologna Abs observations. Fe concentrations were higher than those recommended in the literature, possibly caused by the low mobility and accumulation of Fe up to the end of the plant cycle when the leaves were collected for analysis. Possible contamination due to Fe deposition on the leaf surface, mainly as a result of foliar applications by viticulturists in Italy, may have also played a role.

The APPC7-Estela cultivar is grown for fresh consumption, and fruit quality is therefore crucial for subsistence of the region, which explains why low-yield vineyards can be economically viable. The sufficiency ranges for P, S, Mn, and Zn deemed adequate are higher than those proposed earlier [9], while the levels of the other nutrients are lower than recommended. These variations can be due to the excessive application of phosphate fertilizers, which increases the availability and absorption potential of P, and also of S, a component of phosphate fertilizers [25]. On the other hand, possible contamination with Mn and Zn, as well as with Cu, is associated with the application of pesticides prior to leaf sampling for foliar diagnosis [2,3].

The CLs for the cultivars in the Farroupilha region are close to those recommended [3], but some nutrients such as P in both cultivars and S in the MB cultivar are above the recommended standards [8]. Therefore, it can be assumed that the same standards can be applied to grapevines in the state of São Paulo. A significant characteristic of the crops in Dom Pedrito and Maçambará is the difference in N and K levels deemed adequate in the present study, which are half and twice those suggested [3,8], respectively. This can be attributed, to some extent, to the sandy soils in the Campanha Gaúcha region, which supply less N to the vines, and to the high doses of K applied in the region [8].

Nutrient levels according to CND standards for high-yield populations (mean, standard deviation, and upper and lower bounds) vary as per region and cultivar. The variations in the upper and lower bounds (Equation (7)) indicate that a range around the equilibrium point of the simplex, rather than a single point, should be considered adequate [5,6]. Given that nutritional standards are based on the sampling of high-yield populations, the intrinsic variation in sampling [33] should also be considered as a margin around the simplex equilibrium. For each nutrient, variability should be assessed within the same range as the nutrient itself. Recall that, according to [5,7], adequate nutritional balance should be assessed between the mean and the standard deviation of each multinutrient variable around the simplex.

### 4.3. Recommended Timing for Foliar Diagnosis of the Sangiovese Cultivar in the Emilia-Romagna Region

The paucity of studies on the most appropriate time for assessment of nutritional status of the Sangiovese cultivar grown in the Emilia-Romagna region limits the use of leaf analysis as a method for nutritional evaluation. Variations in mineral composition and their relationship with yield should guide the choice of the appropriate plant tissue [34,35]. In the present study, the significant correlations between nutrient levels and multinutrient variables in relation to yield show similar quantities and importance.

The sampling stages showed similar results between abscissed and Estate (summer) leaves. The multinutrient variables for N, P, K, B, Cu, Zn, and R were significantly correlated in both stages. Calcium and Fe were significant only in the abscissed stage, whereas S and Mn were significant in summer one. The designation of both stages as appropriate, combined with the establishment of standards, allows assessing the nutritional status in both stages, adjusting the fertilization recommendations, as previously suggested [2,3,32].

### 4.4. Implications of the Results Obtained in Viticulture

The findings allow viticulturists to apply region-specific standard protocols to each cultivation area, sampling period, and cultivars. In addition, the established nutritional standards allow for more accurate assessments of nutrient application requirements. In general, in various wine-producing countries around the world [1], nutrient application rates are proposed for all grapevine cultivars, without regional specifications, as suggested in the present study. This is also observed in Brazil [8,9] and in Italy [22], where the assessment of nutritional status of grapevines is proposed at the state and national levels. Nevertheless, this approach is not appropriate, as demonstrated in the present study because each region, each sampling period, or each cultivar has nutritional variations that require more specific accuracy.

More precise determination of fertilizer application requirements for grapevines allows the maintenance of high yields, likely ensuring adequate quality parameters for both the must and wine. A higher yield does not always affect all quality variables of the must, such as total soluble solids, titratable acidity, and pH. On the other hand, a higher yield can reduce the concentration of anthocyanins in the must and in the wine, which might not be desirable, especially for grapevines that naturally produce wines with low color intensity [19]. Furthermore, fertilization applications can be optimized. Caution should be taken with nitrogen and phosphate fertilizers, which are overused in Brazil, mainly in the Campanha Gaúcha southern region, and in the state of São Paulo, in the southeast. This occurs because grapevines are grown in soils with low organic matter (Dom Pedrito and Maçambará) and, consequently, with low availability of N in its mineral form [19], and in soils with high P adsorption capacity (Pilar do Sul), as already described [36]. Potassium fertilizers are used because K is the nutrient that grape clusters take up in the highest amounts [37], but also because some of the applied K can be fixed to the clay minerals of the soil [4,21].

Micronutrient applications, whether by incorporation in soil, fertigation, or foliar spraying, can only be performed when their requirements have been diagnosed. This helps avoiding problems such as complexation and precipitation with macronutrients such as P and Zn, reducing the percentage of free chemical species of these nutrients in the solution and decreasing their uptake by plants [38]. Besides, it will be possible to prevent excessive nutrient availability, including nitrate, which is easily leached out in sandy soils such as those found in the Campanha Gaúcha region [19]. Nitrate leaching significantly increases the contamination of surface waters [39]. In sloped soils, however, nitrate can runoff with surface waters, especially during periods of intense rainfall [19,40].

## 5. Conclusions

Nutritional standards should not be universally applied under any circumstances because they vary across cultivars, regions, and observation periods. Therefore, nutritional standards should be tailored to each situation.

The CND method successfully established critical levels and lower and upper bounds, in addition to nutritional standards, thus contributing to a reference dataset. Therefore, it is suggested that viticulturists implement comprehensive foliar diagnosis and the recommended values to assess the nutritional status of grapevines under local conditions.

The observed similarities in nutritional standards for grapevines indicate that, in some cases, it is possible to extrapolate specific nutritional recommendations for grapevines grown under different conditions, especially when research is still in its early stages. However, the use of universal nutritional standards is discouraged, and introduction of appropriate regional indexes is recommended.

## Figures and Tables

**Figure 1 plants-14-00698-f001:**
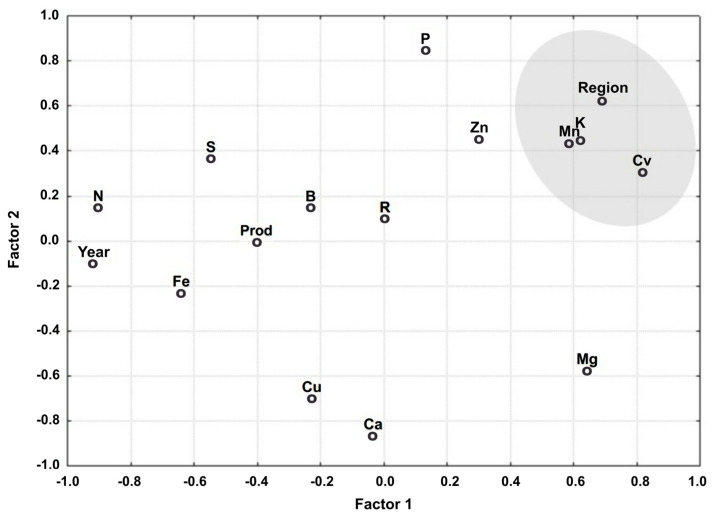
Relationship between the variables associated with Factors 1 and 2.

**Figure 2 plants-14-00698-f002:**
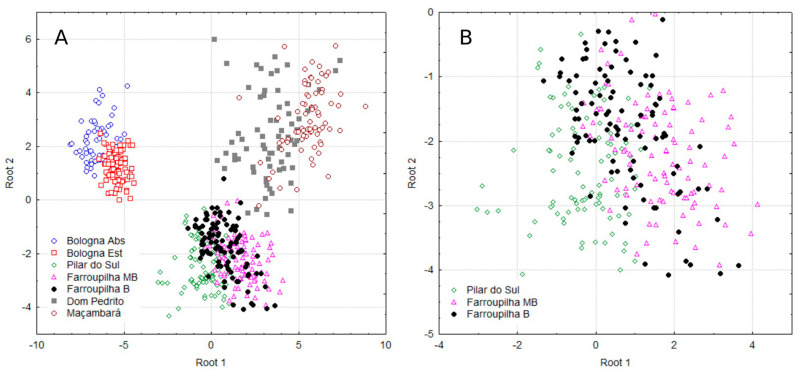
Scatter plot of discriminant analysis for the assessed grapevine population of all groups (**A**) and in Pilar do Sul and Farroupilha for the Moscato Branco (MB) and Bordô (B) cultivars (**B**).

**Table 1 plants-14-00698-t001:** Initial and final sample size after data exclusions, number of observations of the reference population (Ref.), and coefficients of determination (R^2^) for the statistical models examining the relationships between nutrient levels and CND indices in grapevine leaves across regions and cultivars.

Regions	Cultivars	Years of Sample Collection	Observations			R^2^ of Statistical Models										
			Start	End	Ref.	N	P	K	Ca	Mg	S	B	Cu	Fe	Mn	Zn
Bologna	Sangiovese (Abs)	2020–2022	56	53	25	0.81	0.79	0.87	0.57	0.65	0.14	0.80	0.74	0.98	0.51	0.90
	Sangiovese (Est)	2020–2022	96	93	26	0.59	0.58	0.75	0.74	0.82	0.49	0.77	0.98	0.44	0.8	0.98
Pilar do Sul	APPC 7–Estela	2022–2023	95	95	35	0.06	0.70	0.48	0.54	0.67	0.81	0.86	0.98	0.57	0.92	0.91
Farroupilha	Moscato Branco	2020–2021	96	96	60	0.69	0.90	0.56	0.83	0.77	0.93	0.97	0.99	0.65	0.88	0.94
	Bordô	2020–2021	99	99	52	0.22	0.83	0.35	0.98	0.96	0.96	0.97	0.76	0.91	0.86	0.94
Dom Pedrito	Tannat	2006–2016	6	6		0.78	0.91	0.88	0.57	0.90	0.84	0.65	0.96	0.88	0.95	0.87
	Sauvignon Blanc; Chardonnay; Gewurztraminer; Merlot and Pinotage		11	11	33											
	Malbec and Cabernet Sauvignon		4	4												
Maçambará	Cabernet Sauvignon, Merlot and Tannat	2010–2016	6	6	31	0.90	0.94	0.93	0.68	0.83	0.93	0.58	0.92	0.96	0.95	0.93
	Cabernet Franc and Ruby Cabernet		5	5												
	Chardonnay		14	14												
	Malbec, Syrah, Viognier and Pinot Noir		7	7												
	Tempranillo		4	4												

**Table 2 plants-14-00698-t002:** Correlation matrix between the variables and the principal components for the unrotated model and the contribution of each variable to each factor.

Variables	Factor 1	Factor 2	Factor 3	Factor 4
Region	**−0.940**	−0.168	−0.070	0.088
Cultivars	**−0.882**	0.176	−0.027	0.056
Year	**0.837**	−0.406	0.067	0.050
N	0.675	−0.506	−0.089	0.390
P	−0.498	−0.551	−0.433	0.086
K	**−0.826**	−0.040	0.040	0.264
Ca	0.424	**0.731**	0.291	0.156
Mg	−0.309	**0.825**	0.251	0.100
S	0.140	−0.699	0.431	0.373
B	0.316	−0.083	**−0.743**	−0.361
Cu	0.609	0.475	0.018	−0.255
Fe	0.646	−0.143	0.124	0.100
Mn	**−0.793**	−0.201	0.274	−0.088
Zn	−0.481	−0.427	0.174	−0.510
R	0.061	0.220	**−0.737**	0.267
Yield	0.332	−0.407	0.346	−0.382
Expl. Var.	5.905	3.178	1.876	1.106
Prq. Tol.	0.369	0.199	0.117	0.069

Expl. Var. (Explained Variance); Prq. Tol. (Cumulative model full explanation).

**Table 3 plants-14-00698-t003:** Canonical coefficients and discriminant analysis of nutrient levels in the assessment of groups.

Variables	Root 1	Root 2	Root 3	WL ^1^	Root 1	Root 2	Root 3	WL ^1^	Root 1	Root 2	Root 3	Root 4	WL ^1^
	**----------- Regions -----------**				**----------- Cultivars -----------**				**---------------- Year ----------------**				
N	−0.387	−0.662	−0.157	0.002	−0.464	−0.707	−0.611	0.003	0.949	−0.353	0.600	0.774	0.002
P	0.700	−0.224	0.141	0.001	0.623	−0.199	0.133	0.003	0.464	−0.115	−0.118	0.392	0.002
K	0.224	−0.144	0.392	0.001	0.258	−0.041	0.141	0.003	−0.660	−0.458	0.289	1.134	0.002
Ca	−0.781	−0.115	−0.312	0.001	−0.818	−0.201	−0.192	0.003	1.075	−0.114	0.066	0.998	0.002
Mg	0.262	0.574	0.489	0.001	0.258	0.687	0.229	0.003	−0.894	−0.032	−0.603	1.132	0.002
S	0.103	−0.517	0.860	0.002	0.072	−0.432	0.961	0.003	0.322	0.533	−0.930	0.545	0.002
B	−0.641	−0.104	1.058	0.002	−0.639	0.010	1.093	0.003	0.243	−1.086	−0.640	0.910	0.002
Cu	−0.410	−0.599	−0.055	0.001	−0.378	−0.583	−0.119	0.003	0.808	−0.786	0.216	2.333	0.002
Fe	−0.172	−0.391	0.019	0.001	−0.032	−0.485	0.404	0.003	0.458	−0.362	−0.041	1.533	0.002
Mn	0.515	−0.539	−0.481	0.002	0.447	−0.718	−0.328	0.003	0.510	−1.146	0.668	1.307	0.002
Zn	−0.248	−0.247	0.285	0.001	−0.180	−0.177	0.251	0.003	0.244	0.051	−0.398	1.620	0.002
Eigenvalue	14.737	4.929	1.881		11.835	4.967	1.566		13.838	3.087	1.588	1.012	
Cum.Prop.	0.637	0.850	0.931		0.612	0.869	0.950		0.667	0.816	0.892	0.941	
WL **^1^**	0.001	0.017	0.104		0.002	0.029	0.175		0.001	0.017	0.068	0.177	

^1^ Wilks’ Lambda (WL); Cum. Prop. = (cumulative proportion of described differences).

**Table 4 plants-14-00698-t004:** Means of multinutrient CND variables and analysis of variance for the assessment of similarity across the populations regarding variables and regions.

**Variables**	**Bologna Abs**		**Bologna Est**								**Dom Pedrito**			**Maçambará**					
	**---------- Means ----------**				***t*-Value**			*p*			**------- Means -------**					***t*-Value**		*p*	
N	2.620		3.182		−16.210			0.000			2.362			2.063		4.738		0.000	
P	−0.015		0.558		−24.081			0.000			1.095			1.339		−2.645		0.009	
K	1.253		1.784		−10.198			0.000			**2.914**			**2.803**		**1.388**		**0.167**	
Ca	3.459		3.079		12.965			0.000			2.366			2.447		−2.320		0.022	
Mg	1.465		1.244		6.320			0.000			1.624			1.768		−2.180		0.031	
S	−0.068		0.455		−27.118			0.000			0.338			0.078		3.696		0.000	
B	−2.979		−2.630		−7.017			0.000			**−3.532**			**−3.581**		**1.124**		**0.263**	
Cu	−2.058		−2.577		6.194			0.000			−4.127			−4.553		3.712		0.000	
Fe	−1.455		−2.588		19.010			0.000			−3.007			−3.265		2.628		0.010	
Mn	−2.606		−2.995		10.301			0.000			−1.079			−0.155		−9.902		0.000	
**Zn**	**−3.227**		**−3.284**		**0.592**			**0.555**			**−2.516**			**−2.539**		**0.362**		**0.718**	
R	3.611		3.771		−2.599			0.010			**3.562**			**3.596**		**−0.385**		**0.701**	
**Variáveis**	**Pilar do Sul x** **Farroupilha MB**					**Pilar do Sul x** **Farroupilha B**								**Farroupilha MB x** **Farroupilha B**					
	**----- Médias -----**		** *t* ** **-Value**	** *p* **		**----- Médias -----**			** *t* ** **-Value**			** *p* **		**----- Médias -----**			** *t* ** **-Value**		** *p* **
N	3.29	2.92	10.89	0.000		3.29	3.15		6.17			0.000		2.92	3.15		−6.91		0.000
P	0.90	1.08	−3.38	0.001		0.90	1.51		−11.22			0.000		1.08	1.51		−7.14		0.000
K	**2.11**	**2.08**	**0.89**	**0.373**		2.11	2.27		−5.49			0.000		2.08	2.27		−6.05		0.000
Ca	2.45	1.96	9.84	0.000		2.45	1.93		7.40			0.000		**1.96**	**1.93**		**0.39**		**0.695**
Mg	0.59	0.48	2.54	0.012		0.59	0.40		3.16			0.002		**0.48**	**0.40**		**1.21**		**0.227**
S	0.79	1.09	−4.11	0.000		0.79	0.95		−2.31			0.022		**1.09**	**0.95**		**1.62**		**0.107**
B	−3.79	−3.14	−7.41	0.000		−3.79	−2.75		−11.95			0.000		−3.14	−2.75		−3.69		0.000
Cu	−3.12	−4.00	5.07	0.000		−3.12	−4.53		11.18			0.000		−4.00	−4.53		4.40		0.000
Fe	−2.23	−2.60	5.93	0.000		−2.23	−2.06		−2.65			0.009		−2.60	−2.06		−11.10		0.000
Mn	−1.38	−0.82	−8.35	0.000		−1.38	−1.81		6.64			0.000		−0.82	−1.81		17.42		0.000
Zn	−2.98	−2.27	−9.05	0.000		−2.98	−2.69		−3.55			0.000		−2.27	−2.69		4.68		0.000
R	3.38	3.22	3.44	0.001		3.38	3.64		−4.85			0.000		3.22	3.64		−9.62		0.000

Differences between the groups are shown in bold (*p* > 0.05).

**Table 5 plants-14-00698-t005:** Critical level (NC), upper and lower bounds, and parameters for CND standards for the high-yield population, including the mean, standard deviation (SD), and upper (UCB) and lower (LCB) confidence intervals of nutrients for each region and cultivar.

Cultivars/Elements		N	P	K	Ca	Mg	S	B	Cu	Fe	Mn	Zn
Bologna Abs	I_S_+	16.8	1.4	3.9	44.6	5.5	1.0	64.1	130.2	603.2	82.1	69.4
	NC	13.9	1.2	2.9	39.3	4.7	0.9	49.0	109.7	444.2	71.3	56.0
	I_i_−	10.9	1.0	1.9	34.1	3.9	0.8	34.0	89.1	285.2	60.4	42.6
	Mean	2.56	0.08	1.03	3.55	1.48	−0.09	−3.04	−2.26	−1.02	−2.64	−2.97
	SD	0.24	0.16	0.32	0.12	0.23	0.10	0.34	0.27	0.49	0.15	0.21
	UCB	2.69	0.17	1.21	3.61	1.61	−0.04	−2.85	−2.11	−0.74	−2.56	−2.85
	LCB	2.42	0.00	0.85	3.48	1.36	−0.15	−3.23	−2.42	−1.30	−2.73	−3.09
Bologna Est	I_S_+	29.1	2.0	5.8	23.4	4.4	1.8	69.8	156.2	71.2	56.9	56.9
	NC	26.2	1.8	5.0	20.7	3.8	1.6	60.5	110.9	65.2	48.8	40.6
	I_i_−	23.4	1.6	4.2	18.0	3.3	1.4	51.2	65.6	59.2	40.8	24.2
	Mean	3.28	0.61	1.66	3.12	1.37	0.48	−2.77	−2.22	−2.67	−2.98	−3.27
	SD	0.12	0.08	0.24	0.09	0.14	0.11	0.19	0.78	0.10	0.17	0.19
	UCB	3.34	0.66	1.79	3.16	1.45	0.55	−2.67	−1.79	−2.62	−2.89	−3.17
	LCB	3.21	0.57	1.53	3.07	1.30	0.42	−2.87	−2.64	−2.72	−3.07	−3.38
Pilar do Sul	I_S_+	33.1	3.8	11.3	15.1	2.9	3.0	34.8	220.2	142.2	312.5	66.7
	NC	30.4	3.3	10.1	12.8	2.5	2.3	25.5	78.9	116.5	192.8	43.5
	I_i_−	27.7	2.8	8.8	10.6	2.0	1.5	16.2	78.9	90.8	73.2	20.3
	Mean	3.02	0.80	1.87	2.14	0.42	0.42	−4.09	−2.99	−2.53	−2.00	−3.51
	SD	0.14	0.32	0.20	0.21	0.29	0.31	0.43	0.92	0.16	0.56	0.55
	UCB	3.09	0.95	1.97	2.24	0.55	0.57	−3.89	−2.57	−2.46	−1.74	−3.25
	LCB	2.96	0.65	1.78	2.04	0.29	0.28	−4.29	−3.42	−2.60	−2.25	−3.76
Farroupilha Moscato Branco	I_S_+	30.3	5.6	13.4	13.4	2.9	6.7	87.7	96.3	125.2	850.1	257.3
	NC	27.5	4.0	12.1	11.3	2.5	5.0	48.7	21.8	106.3	645.9	158.1
	I_i_−	24.7	2.5	10.8	9.1	2.0	3.4	9.8	21.8	87.4	441.6	59.0
	Mean	2.50	0.48	1.65	1.60	0.10	0.80	−3.83	−4.64	−3.09	−1.27	−2.69
	SD	0.27	0.40	0.28	0.37	0.28	0.59	0.72	1.15	0.27	0.37	0.60
	UCB	2.59	0.62	1.75	1.72	0.20	1.01	−3.58	−4.25	−3.00	−1.15	−2.48
	LCB	2.41	0.35	1.56	1.47	0.01	0.60	−4.08	−5.04	−3.18	−1.40	−2.89
Farroupilha Bordô	I_S_+	25.9	6.0	11.1	12.9	2.8	4.8	93.6	13.6	222.5	217.1	115.2
	NC	23.5	4.7	10.0	9.7	2.1	3.5	58.1	11.1	167.2	165.8	76.3
	I_i_−	21.2	3.4	8.8	6.5	1.4	2.1	22.6	8.5	111.9	114.4	37.4
	Mean	2.65	0.94	1.79	1.68	0.14	0.66	−3.42	−5.06	−2.41	−2.38	−3.18
	SD	0.15	0.43	0.17	0.47	0.43	0.52	0.70	0.30	0.39	0.37	0.65
	UCB	2.70	1.10	1.85	1.85	0.30	0.86	−3.16	−4.95	−2.27	−2.24	−2.94
	LCB	2.59	0.77	1.72	1.51	−0.03	0.47	−3.68	−5.17	−2.56	−2.51	−3.43
Dom Pedrito	I_S_+	10.4	4.5	27.1	15.0	8.4	2.0	37.5	24.9	69.7	605.3	111.6
	NC	7.4	3.1	19.7	12.9	6.1	1.5	32.8	11.5	51.7	399.9	86.6
	I_i_−	4.4	1.7	12.2	10.9	3.9	1.0	28.1	11.5	33.7	194.5	61.5
	Mean	2.40	1.08	2.92	2.42	1.70	0.35	−3.47	−4.49	−3.06	−1.04	−2.52
	SD	0.41	0.54	0.45	0.24	0.53	0.36	0.34	0.57	0.57	0.66	0.39
	UCB	2.63	1.38	3.17	2.55	1.99	0.56	−3.28	−4.17	−2.74	−0.67	−2.30
	LCB	2.17	0.78	2.67	2.29	1.40	0.15	−3.66	−4.81	−3.38	−1.41	−2.73
Maçambará	I_S_+	11.3	5.9	25.0	14.2	8.5	1.8	33.1	15.2	59.0	1401.0	104.2
	NC	8.5	4.5	18.4	12.5	7.0	1.4	28.8	11.8	44.2	980.6	76.8
	I_i_−	5.6	3.0	11.9	10.9	5.6	1.0	24.5	8.3	29.4	560.1	49.4
	Mean	2.01	1.38	2.81	2.43	1.84	0.19	−3.63	−4.56	−3.24	−0.13	−2.66
	SD	0.37	0.53	0.51	0.24	0.30	0.36	0.17	0.42	0.37	0.59	0.39
	UCB	2.19	1.65	3.06	2.54	1.99	0.37	−3.55	−4.36	−3.06	0.16	−2.47
	LCB	1.83	1.12	2.55	2.31	1.69	0.01	−3.72	−4.77	−3.43	−0.42	−2.85
References												
[9]		30–35	2.4–2.9	15–20	13–18	4.8–5.3	3.3–3.8	45–53	18–22	95–105	65–75	30–35
[3]		24–30	2.9–3.8	11–14	12–16	2.6–3.3	3.1–3.8	27–41	10–14	91–142	398–586	148–254
[8]		16–24	1.2–4.0	8–16	16–24	2.0–6.0		30–65		60–150	30–300	25–60
[22]		21–31	1.3–3.1	8–15	16–28	2.0–3.9	1.0–2.3	15–45		60–130	50–220	30–80
[22]-Veraison		18–27	0.9–3.0	7–16	23–39	2.2–4.7	0.9–3.5	16–41		40–220	35–220	10–90
[23]		22.5	1.7	8.4	37.6	4.0						

I_S_+ = upper confidence interval of nutrient content (N, P, K, Ca, Mg, S in g.kg; the B, Cu Fe, Mn, Zn in mg.kg); I_i_− = lower confidence interval of nutrient content (N, P, K, Ca, Mg, S in g.kg; the B, Cu Fe, Mn, Zn in mg.kg); NC = Critical level.

**Table 6 plants-14-00698-t006:** Correlation (r) between yield, nutrient levels, and multinutrient CND standards obtained from leaf blade of Sangiovese grapevines at two developmental stages in Bologna, Emilia-Romagna, Italy.

Stages	N	P	K	Ca	Mg	S	B	Cu	Fe	Mn	Zn	R
Abscissed	−0.12	0.57 *	−0.42 *	0.75 *	0.29 *	0.53 *	−0.05	−0.54 *	0.72 *	0.23	0.77 *	
	−0.37 *	0.51 *	−0.56 *	0.64 *	−0.02	0.01	−0.29 *	−0.71 *	0.79 *	−0.18	0.72 *	−0.53 *
Estate	0.76 *	0.74 *	0.01	0.34 *	0.53 *	0.72 *	−0.24 *	0.61 *	0.03	0.22 *	0.30 *	
	0.28 *	0.34 *	−0.41 *	−0.15	0.18	0.23 *	−0.57 *	0.45 *	−0.57 *	−0.14	0.35 *	−0.77 *

* *p* < 0.05.

## Data Availability

The dataset is available upon request from the corresponding author.

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
