# Peer review of "Proposal of Nutritional Standards for the Assessment of the Nutritional Status of Grapevines in Subtropical and Temperate Regions"

_plants, 2025, doi:10.3390/plants14050698_

Round 1
Reviewer 1 Report
Comments and Suggestions for Authors
The manuscript by Danilo Rozane and colleagues describes a research where different metal concentrations were detected and quantified from vine leaves of different grape varieties collected from 2 Brazilian regions and from 1 Italian wine region.
The topic is interesting from a scientific and technical point of view. The data obtained could be useful for a better understanding the potential influence of the specific location of the vineyards studied on leaf vine metal composition. However, several major comments could be considerate:
1 - This research and results will only be valid for the specific locations where the samples were collected. Thus, the title of the article is quite speculative (“….universal guidelines….”), and is even contradictory to some parts of the text written by the authors (example: 449-450 and all the conclusions).
2 - The abstract must be fully written, as it is not clear what the results obtained were and the conclusions that can be drawn from this work.
3 - The introduction should be more in-depth to reveal the state of the art of the topic studied and also to demonstrate the innovation of the work in this article.
4 - It must be clarified if the grape varieties studied are used to produce table grapes or wine.
5 – Line 69-70 “Vitis vinifera …” must be in italic form. Revise in all manuscript.
6 - The age of the vines, soil characteristics, type of fertilizations carried out on the vineyards 1 to 2 years before the sample collection, and climate data are several of the parameters that must be presented, as they influence the results obtained.
7 – Line 454-455: This is not totally true, because wines with low anthocyanin content and red color It will not be an obvious aspect of low quality, considering current market trends.
Considering all these comments, I recommend that the article should be greatly improved, as the way it is presented is not entirely innovative.
Author Response
Dear Editor and Reviewers,
We are grateful for the opportunity to review our manuscript entitled "Proposal of nutritional standards for the assessment of the nutritional status of grapevines in subtropical and temperate regions". The reviewers' suggestions were essential to improving the quality of the work.
Below, we respond to each point raised:
Reviewer 1
The manuscript by Danilo Rozane and colleagues describes a research where different metal concentrations were detected and quantified from vine leaves of different grape varieties collected from 2 Brazilian regions and from 1 Italian wine region.
The topic is interesting from a scientific and technical point of view. The data obtained could be useful for a better understanding the potential influence of the specific location of the vineyards studied on leaf vine metal composition.
Reply: We appreciate your comments and thank you for contributing to the improvement of the manuscript.
However, several major comments could be considerate:
1 - This research and results will only be valid for the specific locations where the samples were collected. Thus, the title of the article is quite speculative (“….universal guidelines….”), and is even contradictory to some parts of the text written by the authors (example: 449-450 and all the conclusions).
Reply: We appreciate your observation, and the title has been changed.
2 - The abstract must be fully written, as it is not clear what the results obtained were and the conclusions that can be drawn from this work.
Reply: We appreciate your feedback and have edited the abstract as requested by you and the other reviewer.
3 - The introduction should be more in-depth to reveal the state of the art of the topic studied and also to demonstrate the innovation of the work in this article.
Reply: We appreciate your feedback and have added three paragraphs to the introduction with the information you and the other reviewer requested.
4 - It must be clarified if the grape varieties studied are used to produce table grapes or wine.
Reply: We have adjusted the text to make it clear that the study included 18 grape varieties used in the production of wine and sparkling wine from five different locations (line. 90-91)
5 – Line 69-70 “Vitis vinifera …” must be in italic form. Revise in all manuscript.
Reply: Done.
6 - The age of the vines, soil characteristics, type of fertilizations carried out on the vineyards 1 to 2 years before the sample collection, and climate data are several of the parameters that must be presented, as they influence the results obtained.
Reply: We accepted your suggestion and added information in text form (line 109-129) and in table form in the appendix (Table A1).
7 – Line 454-455: This is not totally true, because wines with low anthocyanin content and red color It will not be an obvious aspect of low quality, considering current market trends.
Reply: We appreciate your comments. Anthocyanin content is one of the variables analyzed and normally associated with must and wine quality (Stefanello et al., 2021). Wines with a higher color may be accepted by the consumer market, as well as wines with a lower color intensity. However, in most cases it is not desirable for management practices to reduce anthocyanin levels in must and wine, especially in cultivars where anthocyanin levels are naturally lower.
Stefanello, L. O., Schwalbert, R., Schwalbert, R. A., Drescher, G. L., De Conti, L., Pott, L. P., Tassinari, A., Kulmann, M. S. de S., da Silva, I. C. B., & Brunetto, G. (2021). Ideal nitrogen concentration in leaves for the production of high-quality grapes cv ‘Alicante Bouschet’ (Vitis vinifera L.) subjected to modes of application and nitrogen doses. European Journal of Agronomy, 123, 126200. https://doi.org/10.1016/j.eja.2020.126200
Reviewer 2 Report
Comments and Suggestions for Authors The topic of the manuscript is the nutrient supply of grapes, for which the authors carried out measurements in different landscape units. Research on adequate nutrient supply is very important for sustainable viticulture, especially in different areas. The CND method can be suitable for this purpose. The nutrient requirements of grapes is a long researched and richly published topic, this manuscript is highlighted because of CND. The paper is well written, the English language is understandable, the editing is neat and the chapters are well written. The text of the manuscript is easy to understand and does not suffer from grammatical and spelling errors. The conclusions are consistent with the evidence and arguments presented. The method used is appropriate to the questions presented. The topic is well circumscribed and clear. The main question is answered by the authors. There are suggestions for modifications: I suggest rewriting the abstract so that it currently contains the problem statement, background knowledge, and importance of the topic. Then the hypothesis, main results with figures and the conclusion. Introduction: I propose to expand the section on the nutrient uptake of grapes and to include literature data on the CND method.
Author Response
Dear Editor and Reviewers,
We are grateful for the opportunity to review our manuscript entitled "Proposal of nutritional standards for the assessment of the nutritional status of grapevines in subtropical and temperate regions". The reviewers' suggestions were essential to improving the quality of the work.
Below, we respond to each point raised:
Reviewer 2
The topic of the manuscript is the nutrient supply of grapes, for which the authors carried out measurements in different landscape units. Research on adequate nutrient supply is very important for sustainable viticulture, especially in different areas. The CND method can be suitable for this purpose. The nutrient requirements of grapes is a long researched and richly published topic, this manuscript is highlighted because of CND. The paper is well written, the English language is understandable, the editing is neat and the chapters are well written. The text of the manuscript is easy to understand and does not suffer from grammatical and spelling errors. The conclusions are consistent with the evidence and arguments presented. The method used is appropriate to the questions presented. The topic is well circumscribed and clear. The main question is answered by the authors.
Reply: We appreciate your comments and thank you for contributing to the improvement of the manuscript.
There are suggestions for modifications: I suggest rewriting the abstract so that it currently contains the problem statement, background knowledge, and importance of the topic. Then the hypothesis, main results with figures and the conclusion.
Reply: We appreciate your feedback and have edited the abstract as requested by you and the other reviewer.
Introduction: I propose to expand the section on the nutrient uptake of grapes and to include literature data on the CND method.
Reply: We appreciate your feedback and have added three paragraphs to the introduction with the information you and the other reviewer requested.
Round 2
Reviewer 1 Report
Comments and Suggestions for Authors
The paper has been revised and greatly improved. Overall, the reviewer's comments have been taken into account. Therefore, I have no further comments to consider.
If the Editor of the Journal decides to accept the article, I have no objections.
Reviewer 2 Report
Comments and Suggestions for Authors
I recommand it for publication